# Regional Renewable Energy Installation Optimization Strategies with Renewable Portfolio Standards in China

**Yuanyuan He, Luxin Wan, Manli Zhang and Huijuan Zhao \***

School of Economics and Management, Qingdao University of Science and Technology, Qingdao 266061, China
* Correspondence: zhj.smart@126.com

**Abstract:** In this paper, we provide theoretical and policy support for quota-allocation strategies based on a national unified renewable energy (RE) power market. Renewable portfolio standards (RPSs) are of great significance in promoting the stable development of renewable energy and improving power market decision making in China's power industry. To resolve the geographical, resource allocation, and power-grid problems of multi-regional RE power generation, we constructed a regional distribution optimization model with the lowest cost under the RPS policy and designed a set of dynamic distribution mechanisms based on the renewable energy power quota index. The results show that it is necessary to prioritize development of wind-generated power on the North China and Northeast Power Grids, solar energy on the Northwest Power Grid, and biomass energy generation on grids in other regions to plan specific task undertakings and allocate RE power generation to each grid. We propose a multi-regional power distribution model at the lowest cost under the RPS policy to provide solutions and references for renewable energy power market quota allocation.

**Keywords:** renewable portfolio standard; renewable energy; installation optimization strategies; optimization model; power grid

## 1. Introduction

The rapid development of social economies has led to increased industrial production and energy consumption. However, fossil fuels—-coal, oil, and natural gas—-are unsustainable and rapidly running out; moreover, their extensive use generates carbon dioxide emissions that contribute to global warming. At the United Nations General Assembly in September 2020, China announced it would adopt more powerful policies to achieve peak carbon dioxide emissions by 2030 and carbon neutrality by 2060. Therefore, optimizing an energy structure dominated by new energy and promoting grid-connected consumption of renewable energy on a large scale are important for solving the global energy crisis, addressing climate change, and achieving China's medium and long-term goals of "peak carbon dioxide emissions and carbon neutrality".

Emerging industries that use photovoltaic cells, wind power, and other types of new energy cannot develop without government subsidies in their early stages of development; however, subsidies do not last long. As the renewable energy industry develops, the support and subsidy mechanism will be phased out, and the energy industry will enter the commercialization stage. Finally, a policy system combining government guidance and a green power market mechanism will form.

Implementation of the renewable portfolio standards (RPS) proposed in China's 12th Five-Year Plan has become a powerful guarantee for solving the problems of power generation, grid establishment, and renewable energy consumption. It minimizes government subsidies and encourages the development of renewable energy through market-oriented means and has facilitated the promotion renewable energy development in countries that implement a quota system. China's power-market reform started relatively late, and because renewable energy is in a dire state, it is necessary to find ways to promote its

development. China is only in the early stages of its 14th Five-Year Plan, so to speed up the marketization and application of renewable energy, implementing the RPS policy was inevitable. As an effective tool for promoting power structure reform, it is an important part of the supply-side reform of an energy structure.

In recent years, issues related to the RPS policy have become the focus of research in the energy and environmental policies. Research topics include the formulation and implementation of the RPS policy and resource allocation ([1–7]). Bhattacharya et al. [8] analyzed changes in market welfare after the introduction of the RPS policy and explained the effect of a quota system on supply and demand. The results show that energy prices rose after the implementation, that policies introducing a renewable energy quota system played a key factor, and that some seemingly best cases led to a decline in total renewable energy consumption. Bao et al. [9] simulated the behavior of buyers and sellers in a market with the RPS policy and quota trading using the evolutionary game model. The analysis showed that quota trading not only improved the price competitiveness of renewable energy power-generation enterprises but also reduced the profits of fossil-fuel power-generation enterprises. Additionally, when the quota standard was raised, the state power grid benefited from the difference between the lower wholesale price of thermal power and the constant retail price. Zhao et al. [10] used the evolutionary game model to simulate the behavior of energy producers, analyze the symbiotic evolution of the RPS policy and enterprise behavior strategy, and discuss the influence of key parameters on the evolution of power producers. The results show that a higher unit penalty, a lower transaction cost, and a marginal cost difference between clean and traditional energy could improve the efficiency of a renewable energy quota system with an optimal quota. Xu et al. [11] established a quota allocation model based on the lowest renewable energy cost; determined the renewable energy production target (LC-RPS) of 30 provinces; constructed an index system based on energy consumption, resource endowment, historical responsibility, and transmission facilities; determined the RPS target (IN-RPS); and compared the above two scenarios with the RPS target allocation scenario (GM-RPS) and NO-RPS scenario published by the government. Yu et al. [12] established a renewable energy power-dispatching model, proposed an economically feasible strategy of dispatching power from renewable energy in China's provinces from 2020 to 2022 to meet the requirements of the RPS policy, assessed the corresponding pressure of each province, and finally put forth solutions for each province to achieve the RPS goal. Wang et al. [13] developed a mid-to long-term optimization model for the area served by the China Southern Power Grid Corporation (CSPGC) based on power plant constraints and the RPS goals for 2016–2030. The results provide an optimal method for the five provinces in the CSPGC area to achieve their RPS goals.

Implementing the RPS has had a far-reaching impact on the regional power industry. Unlike foreign studies on a renewable energy quota system, which started earlier, China's has no implementation experience, and it lacks renewable energy quota research. This paper studies the multi-regional RE power allocation model of China's power industry that has the lowest cost under the RPS policy. Considering the geographical location, resource endowment, and power grid structure, power transmission between regions can meet the demand of each region. A dynamic allocation mechanism of a renewable energy power quota index is designed to provide theoretical and policy support for a national unified renewable energy power market. There are two main contributions of this study. The first is to meet regional RPS target constraints in case of increasing power demand, to establish a regional renewable energy generation optimal allocation model, and to design a dynamic allocation strategy for each grid to achieve RPS medium and long-term goals at the lowest cost. Compared with the existing literature, this model provides a more detailed RE generation configuration scheme with higher temporal and spatial accuracy. The second is the design of three scenarios: optimistic, neutral, and pessimistic. Comparing the power allocation schemes and exploring the implementation paths of the renewable energy quota systems in different scenarios and analyzing the impact of COVID-19 on the

implementation of RPS policies in different regions will ensure higher energy efficiency in each region.

The remainder of the paper is organized as follows. Section 2 presents a literature review. Section 3 establishes the regional power installation optimization model. Section 4 designs an algorithm. Section 5 analyses data sources and scenario settings. Section 6 provides results and discussions. Section 7 concludes the paper.

## 2. Literature Review

Research into the optimization of renewable energy installations has several aspects. First, is the impact of RPS policy on the renewable energy power industry. By 2060, China intends to achieve carbon neutrality and will have implemented the renewable energy portfolio standards; therefore, Zhou et al. [14]), based on equality and efficiency as well as a zero and gain data envelopment analysis and entropy model, promote the development of renewable energy management by distributing China's renewable energy based on a provincial quota. Uncertainty has an important impact on the production of renewable energy, so Yang et al. (2021) [15] compared the influence of a feed-in tariff and renewable energy portfolio standard in developing a renewable energy industry under uncertainty through a two-stage model. Song et al. (2021) [16] explored the influence of policy parameters such as RPS quota target planning, unit penalty, and TGC price cap to construct a system-dynamics model of a multi-market-coupled trading system involving a renewable power market and an over-quota consumption market based on the newly issued RPS policy. It found that a new RPS not only affects prices and trading volumes in multiple markets but also boosts renewable energy generation. Joshi [17] assessed the impact of renewable energy portfolio standards on renewable power capacity. Yu et al. [12] optimized the inter-provincial renewable energy dispatching strategy to meet the requirements of renewable energy generation. They established the renewable energy power dispatching model and concluded the economic and feasible strategy of renewable energy power dispatching in China's provinces from 2020 to 2022.

Secondly, most scholars' research focuses on optimizing renewable energy systems. For example, Liu et al. [18] provide a game-theory-based modeling framework and customized solution strategies for optimizing multi-energy system design and renewable energy subsidy strategies. Cosic et al. [19] found that renewable energy communities can reduce total energy costs by 15% and total $CO_2$ emissions by 34% through optimized selection and operation of energy technologies. Fan et al. [20] established a comprehensive planning model composed of multiple regression and linear programming models based on the proposed 2020 provincial renewable energy portfolio standard. Through the inter-provincial allocation of renewable energy and the combined renewable energy portfolio standard and green certificate trading system, they promoted renewable energy resource configuration optimization. One of the ways to improve power reliability is to combine multiple renewable energy and storage systems. Memon and Patel [21] provide a comprehensive overview of the different scale and optimization methods developed by the research community. Considering the availability of renewable energy resources, Makhloufi et al. [22] adopted the multi-objective cuckoo search algorithm to solve the multi-optimization problem of energy strategies. Deveci and Güler [23] proposed a two-step multi-objective optimization framework for renewable energy planning, taking into account different situations of renewable energy investment expenditure and optimal use of resource availability. When developing sustainable local energy systems, Hori et al. [24] visualized pareto solutions for the optimal renewable energy mix using an environmentally sustainable renewable energy region optimization utility with a multi-objective evolutionary algorithm. The use of hybrid renewable energy systems holds great promise for sustainable electrification and supporting countries to achieve their energy access goals (Elkadeem et al. [25]). The use of optimized multi-energy systems, including renewable energy, cogeneration, and energy storage, has been shown to be effective in reducing carbon emissions (Martelli et al. [26]). Liu et al. [27] improved the planning of distributed energy systems by adopting an in-

tegrated optimization method that considered equipment configuration and operation strategy. Appropriately optimized control methods are important for ensuring efficient, safe, and high-quality power transmission (Hannan et al. [28]). Regional energy systems are based on the advantages of different types of energy. Lei et al. [29] construct an economic and efficient regionally integrated energy system for scenario tree path optimization considering long-term multiple uncertainties. Sobhani et al. [30] proposed a future-oriented optimization approach for renewable energy to adapt to climate change and energy price changes. The optimal design of a regionally integrated energy system is necessary to realize the effective use of various energy resources and improve energy efficiency and economic benefits. Li et al. [31], based on typical residential and commercial modules constructed on actual regional blocks, studied the capacity allocation optimization of integrated energy and configured multi-energy systems and optimized real-time calculation results. Zhou and Zhou [32] proposed an integrated imprecise optimization framework and discussed their potential applications in hybrid renewable energy systems. The main challenges of implementing renewable energy systems include geographic constraints on resources and technologies, increased energy and material demand, and the need to reduce emissions while remaining cost-sensitive. Kakodkar et al. [33] analyzed the conversion of renewable energy systems and reviewed optimization methods to provide a reference for addressing the challenges.

Power fluctuation is a key problem for the wide adoption of renewable energy grid-connected microgrids. Li et al. [34] proposed a rolling optimization strategy for microgrids that considers the smoothness of grid-connected power fluctuation to solve the grid-connected power fluctuation problem of microgrids. Wei et al. [35] proposed a power balance control method for a renewable-energy-integrated power system based on deep reinforcement learning to optimize a reasonable usage rate. Even the extensive use of renewable energy does not guarantee increased sustainability throughout the planning period (Atabaki et al. [36]). Moreover, renewable energy is intermittent and energy demand is uncertain. Therefore, Liu et al. [37] proposed a new method of power optimization of a tie line to solve this problem. Al-Shahri et al. [38] described the latest intelligent optimization methods in solar energy systems, including their functions, constraints, research gaps, and contributions. The main objective of the optimization method is to reduce investment, operating and maintenance costs, and emissions to improve system reliability. Tang et al. [39] proposed a quota proportion allocation strategy based on game theory to ensure the effective implementation of a renewable energy absorption mechanism. Vitor and Vieira [40] proposed a new voltage optimization evolution system to solve the multi-objective optimization problem. The proposed mathematical model allows the development of specialized search mechanisms to provide more realistic and cost-saving solutions for distribution system operators.

At the same time, many scholars have established various optimization models to eliminate the bottleneck of renewable energy power transmission. For example, Ding and Wei [41] proposed a two-layer optimization model with operational feedback under the demand–response scenario to determine the planning scheme of a regional energy system, minimize total cost, and achieve optimal operational benefits. Potrč et al. [42] introduced an integrated 27-country sustainable renewable energy supply network in the European Union and developed a multi-cycle mixed-integer programming model, in which wind farms have proven to be the most promising solution for the rapid expansion of renewable energy generation.

Ogunmodede et al. [43] demonstrated the ability of integer programming optimization models to minimize capital, operating, and utility costs while optimizing corresponding scheduling strategies. To eliminate the bottleneck of renewable energy power transmission and improve cross-regional absorption capacity, Yu et al. [44] established a multi-objective optimization model for transmission line layout considering grid stability and resource elasticity. Hou et al. [45] developed a new multi-level stochastic mixed-integer model that uses a two-stage acceleration method to solve the two key consequences of complex uncer-

tainties in variable renewable energy when planning power transition, as well as the need to coordinate extensive transmission investment with newly installed power generation facilities. Cho et al. [46] used mixed integer linear programming to develop multi-site and multi-period optimization models to determine the best investment timing and regional allocation and to select the appropriate biomass types and technology mix. Daraei et al. [47] developed an optimization model for regional energy systems to evaluate the impact of local renewable energy systems on production planning for regional cogeneration plants. The sustainable global transition to low-carbon energy systems has given rise to clean power systems that integrate a higher proportion of renewable energy. Existing studies have taken this restructuring process into account. Deng and Lü [48] studied in depth the changes in optimization models brought about by the massive penetration of variable renewable energy by comparing traditional planning models. Li et al. [49] proposed a long-term multi-regional power system planning model that describes the fluctuations of renewable energy; the optimization of installed capacity, power grid, and storage facilities; and resolves the policy uncertainties of carbon tax and power substitution. A large number of optimization methods in renewable energy applications have been widely used to help decision making to reduce computational constraints. Zakaria et al. [50] comprehensively reviewed the general steps of stochastic optimization in renewable energy applications in uncertainty modeling and relevant information sampling, respectively. The research on stochastic optimization methods in renewable energy applications was reviewed and the related future research fields were identified. She et al. [51] optimized the energy cost and the original design of the energy optimization model with multiple objectives by enhancing the Archimedes optimization algorithm, optimized it by using the mixed gray multi-level comprehensive evaluation method and provided the optimal solution through the selection decision based on a global optimal model.

Since energy storage can improve the usage rate of renewable energy, Wang et al. [52] established a capacity expansion planning model that considered multi-function hybrid energy storage and proposed an operational strategy in which hybrid energy storage participates in the demand response, thus improving the calculation accuracy of the annual cost of energy storage. Yang et al. [53], considering the correlation between wind power and photovoltaic power generation and load, proposed a two-layer optimization operation model of the power market, established a probability model for flexibility requirements, and established a two-layer optimization model for multiple markets to promote the absorption of renewable energy and improve the flexibility of the power system. Li et al. [54] proposed a two-level robust game model to improve the economy and scheduling flexibility of the regionally integrated energy system with large-scale centralized renewable energy, providing theoretical guidance for the optimal scheduling of regionally integrated energy systems and large-scale renewable energy. The existing reconfiguration grid and the adoption of renewable energy power generation expansion costs significantly reduce greenhouse gas emissions, and Bayatloo and Bozorgi-Amiri [55] provide a new extended ECCN to solve the dynamic optimization model of environment design and planning by considering the cost reduction in the net present value and improving the efficiency of the network to meet demand. To extract the variation characteristics of power demand effectively, Niu et al. [56] used the Markov chain model to improve prediction accuracy based on interval optimization, accurately predict the medium-term power demand, and ensure the stable and efficient operation of the power system. Yang et al. [57] proposed a distributed robust optimal scheduling model to minimize carbon emissions in the system and abandon traditional energy power generation to promote the realization of carbon peak and carbon neutrality in the power industry.

The above research discussed the optimization of renewable energy devices and is committed to solving the shortcomings of the renewable energy system and optimization of power distribution to achieve sustainable development of renewable energy. However, most studies did not consider RPS. This paper proposes a multi-regional power distribution model with the lowest cost for China's power industry under the RPS policy that provides

solutions and references for renewable energy power market quota allocation schemes. The analysis of the above problems can not only grasp the impact of RPS policy on the power industry but also provides powerful theoretical methods and solutions for the formulation of China's renewable energy policy and planning cross-regional transmission lines.

### 3. Regional Power Installation Optimization Model

*3.1. Symbol Definition (Nomenclature)*

3.1.1. Subscript

$I$: Collection of renewable energy types, $I = \{1, 2, \cdots, n\}, i \in I$.
$J$: Collection of types of disposable energy, $J = \{1, 2, \cdots, m\}, j \in J$.
$K$: Collection of grid area, $K = \{1, 2, \cdots, s\}, k \in K$.
$T$: Set of years within the planning time range, $T = \{1, 2, \cdots, P\}, t \in T$.

3.1.2. Parameters

$C_{ikt}$: Unit levelized cost of renewable energy $i$ of power grid area $k$ in $t$ year.
$h_{ikt}$: Annual power generation duration of renewable energy $i$ of power grid area $k$ in $t$ year.
$PTC_{kk'}$: Unit transmission cost between power grids.
$UL_{ikt}$: Upper limit of the development resources of the energy of power grid area $k$ in $t$ year.
$EF_j$: $CO_2$ emission coefficient of thermal power generation of disposable energy $j$;
$D_{kt}$: Power demand of power grid area $k$ in $t$ year.
$PTL_{kk'}$: Inter-grid transmission loss rate.
$PRE_{kt}$: Proportion of renewable energy of power grid area $k$ in $t$ year.
$TP_{jkt}$: Thermal power generation capacity of disposable energy $j$ of power grid area $k$ in $t$ year.
$f_i / f_j / f_d$: Flexibility coefficients.

3.1.3. Variables

$IC_{ikt}$: Installed capacity of renewable energy $i$ of power grid area $k$ in $t$ year.
$PT_{ikk't}$: Power transmission between renewable energy $i$ in $t$ year ($k \rightarrow k'$) (Transmission from power grid area $k$ to power grid area $k'$).

*3.2. Objective Function*

In the multispecies RE power optimal allocation model under the RPS policy, the objective function is the total cost of all power industries in China, mainly production and transmission cost. The total target cost is obtained by summing the power production cost of different time series, grid regions, power generation forms, and transmission cost between different power grids.

The power generation cost is mainly the initial investment and annual cost. Among them, the initial cost is the one-time investment cost of the equipment, and the annual cost includes operation and maintenance costs, interest payments, and taxes. According to the learning curve theory, the equipment production cost decreases as the equipment output and R&D funds increase [58]. Therefore, the unit investment cost $C_{ni}$ based on the two-factor learning curve can be expressed as:

$$C_{ni} = A_{ic} \times Q_{ni}^{-\alpha_i} \times R_{ni}^{-\beta_i} \tag{1}$$

where, $A_{ic}$ is the unit investment cost of energy $i$ in the base year; $Q_{ni}$ is the cumulative installed capacity; $R_{ni}$ is the accumulated R&D funds; $\alpha_i$ is the learning rate index of $Q_{ni}$; $\beta_i$ is the value range of $R_{ni}$ and the value range of $\alpha_i$; and $\beta_i$ is (0, 1).

The levelized power cost and annual cost of renewable power are, respectively, expressed as:

$$LCOE = \frac{C_{ni} \times S_i + \sum\limits_{n=0}^{N_i} \left( Annual \cos t_{ni} \times (1+r)^{-n} \right)}{\sum\limits_{n=0}^{N_i} \left( S_i \times H_{ni} \times (1+r)^{-n} \right)} \tag{2}$$

$$Annual \cos t_{ni} = O\&M_{ni} + INT_{ni} + TAX_{ni} \tag{3}$$

where, $i$ is the type of renewable energy; $FIT_i$ is the feed-in tariff of renewable energy $i$; $N_i$ is the life cycle of the equipment $i$; $S_i$ is the rated capacity of the equipment $i$; $H_{ni}$ is the actual operating hours of the equipment $i$ in the year $n$; $r$ is the discount rate; $O\&M_{ni}$ is operation and maintenance costs; $INT_{ni}$ is interest expense; and $TAX_{ni}$ is taxes.

Power transmission may exist between different power grids, and the transmission distance between regions is relatively long, usually more than 600 km. Therefore, high voltage transmission is adopted. Equation (4) represents the total transmission cost including high-voltage transmission.

$$LTC = \sum_{t=t_0}^{P} \sum_{k=1}^{s} \sum_{k'=1}^{s} \frac{\sum_{i=1}^{4} PTC_{kk'} \times PT_{ikk't}}{(1+r)^{t-t_0}} \tag{4}$$

Therefore, the objective function is the RE generation cost and interregional transmission cost accumulated according to the time series.

$$TC = \sum_{t=t_0}^{P} \sum_{k=1}^{s} \sum_{i=1}^{n} \frac{C_{ikt} \times IC_{ikt} \times h_{ikt}}{(1+r)^{t-t_0}} + \sum_{t=t_0}^{P} \sum_{k=1}^{s} \sum_{k'=1}^{s} \frac{\sum_{i=1}^{4} PTC_{kk'} \times PT_{ikk't}}{(1+r)^{t-t_0}} \tag{5}$$

### 3.3. Constraints

3.3.1. RE Resource Potential Constraint

China has a vast territory, and the resource endowments of various regions are different, resulting in different available RE consumption of each power grid and an upper limit for the development of non-fossil energy. The maximum potential of the installed capacity of each power grid is shown in Equation (6).

$$IC_{ikt} \times h_{ikt} \leq UL_{ikt} \tag{6}$$

3.3.2. Regional RPS Target Constraint

According to China's RE energy structure and RPS policy, the proportion of the total renewable energy power used in the power grid area (including the renewable energy power produced and imported in the region) to the total annual power demand should reach a minimum standard, that is, the regional RPS targets' constraint. In addition, the transmission loss among regions should be considered. Therefore, the target constraints of regional RPS quota allocation are as follows:

$$\sum_{i=1}^{n} IC_{ikt} \times h_{ikt} + \sum_{k=1}^{s} \left\{ \sum_{i=1}^{n} PT_{ikk't} \times (1 - PTL_{kk'}) - \sum_{i=1}^{n} PT_{ik'kt} \right\} \geq D_{kt} \times PRE_{kt} \tag{7}$$

The power demand $D_{kt}$ of a power grid is affected by the local economic development level (GDP),

$$D_{kt} = MGDP_{kt}(1 + r_{GDP,k,t})q_{kt}(1 - r_{q,GDP,k,t}) \tag{8}$$

where, $MGDP_{kt}$ is the GDP of the power grid area $k$ in $t$ year, $q_{kt}$ is the power demand per unit of GDP, $r_{GDP,k,t}$ is the GDP growth rate of the power grid area $k$ in $t$ year, and $r_{q,GDP,k,t}$ is the decline rate of power demand per unit GDP of the power grid area $k$ in $t$ year.

### 3.3.3. Constraints on Inter-Regional Transmission

The total amount of export transmission in the area where each power grid is located cannot exceed the total amount of power generated by other technologies in the area and transmitted by other areas (excluding transmission loss). The transmission constraints of different types of power are shown in Equation (9).

$$IC_{ikt} \times h_{ikt} + \sum_{k=1}^{s} PT_{ik'kt} \times (1 - PTL_{k'k}) \geq \sum_{k=1}^{s} PT_{ikk't} \qquad (9)$$

### 3.3.4. Flexibility Constraint

Traditional power system models are composed of rigid constraints. With the development of power systems, the influence of uncertainty increases. The flexible method in the power system is mainly the response method for various uncertain factors. Therefore, this paper uses the flexibility index to reflect the strength and possibility of system uncertainty. The specific flexible treatment method is shown in Equation (10).

$$\sum_{i=1}^{n} IC_{ikt} \times h_{ik} \times f_i + \sum_{j=1}^{m} TP_{jkt} \times f_j + D_{kt} \times f_d \geq 0 \qquad (10)$$

## 4. Algorithm Design

Based on the multispecies RE power optimal allocation model under the RPS policy constructed in the previous part, this paper proposes an improved genetic algorithm (IGA) to solve the model. The specific steps are as follows:

**Step 1** Chromosome Coding. The real number coding method, including multiple regional power grids and multiple power types, is used, and the cell structure storage is adopted. The storage structure is $m \times n$. Chromosomes are expressed as:

$$X = \begin{pmatrix} x_{11} & x_{12} & \cdots & x_{1n} \\ x_{21} & x_{22} & \cdots & x_{2n} \\ \cdots & \cdots & \cdots & \cdots \\ x_{m1} & x_{m2} & \cdots & x_{mn} \end{pmatrix}$$

Additionally, the row of the matrix represents the power grid area, and the list shows the power type.

**Step 2** Initial Population and Fitness Calculation. To improve the global search performance and the quality of the solution of the genetic algorithm, the initial random number is transformed to make it between 0 and the maximum potential of the installed capacity, and the chromosomes satisfying the constraint Equations (7), (9) and (10), which are retained as the initial population. Each chromosome is brought into the objective function, and $f(x) = 1/TC$ is defined as the fitness function to place the dominant individual in the dominant set.

**Step 3** Choice. The new species group is selected by the roulette method, and the dominant individuals obtained in step 3 are added to the new species group to improve the population quality.

**Step 4** Crossing. According to the crossover probability, the parents participating in the crossover operation are selected from the new species group, and they are randomly paired to cross to produce offspring. Due to the exchange of some genes in the cross operation, it is difficult for some offspring to meet the constraints. Therefore, it is necessary to conduct a feasibility test on the chromosomes of the offspring, that is, to test whether each gene meets the constraint Equations (7)–(10).

**Step 5** Variation. To improve the population diversity, chromosomes were randomly selected from the new species group by mutation probability. The specific operations are as follows: (1) Randomly select the genes used for mutation operation and exchange them,

and (2) Carry out the feasibility test on the variation, i.e., check whether each gene meets the constraint Equations (7), (9) and (10).

   **Step 6** Termination Discrimination. If the algorithm does not reach the maximum number of iterations, return to step 3. Otherwise, enter the stage of advantage set selection.

   **Step 7** Advantage Set Selection. Through the calculation of applicability, the dominant individuals of various groups are retained to form a dominant set. After the termination of the algorithm, the dominant set is screened to obtain the optimal individual.

*Algorithm Performance Analysis*

   Before solving the generation allocation optimization model, set the parameters for the genetic algorithm: the initial population size is 500; the crossover probability is 0.6; the mutation probability is 0.02; and the maximum evolution number is 250. The working software is Python 3.7 and the computing environment is Intel (R) core (TM) i7_7500u CPU @ 2.70 GHz, 16 GB operating memory, and Windows10 operating system.

   Then, analyze the solution of the improved genetic algorithm. Table 1 covers the algorithm performance of calculating the optimal target value after normalization, showing that the target mean, optimal, and worst values, and the average deviation is better when IgA solves the RE power distribution problem. The performance analysis shows that IGA can not only ensure the convergence speed but also improve the quality of solutions when solving multi-objective problems.

**Table 1.** Algorithm Performance Analysis for Solving Optimization Problems.

| Algorithm | Target Mean Value | Optimal Value | Worst Value | Average Deviation |
|:---:|:---:|:---:|:---:|:---:|
| IGA | 0.278 | 0.214 | 0.354 | 0.020 |

## 5. Data Sources and Scenarios Setting

### 5.1. Scenario Design

   This study designs three scenarios: optimistic, neutral, and pessimistic. The optimistic scenario means that the global pandemic has been effectively controlled; the duration of COVID-19 in some places is relatively short; the impact on the economy is limited to the service industry; and the economic recovery will be relatively happy. The neutral scenario refers to an increase in the duration of the pandemic and an expansion in the impact on major economies from the service industry to manufacturing. The pessimistic scenario describes the spread of the pandemic, affecting global economic development, and an economic impact lasting for three months. Based upon the calculation of global GDP growth by the China Institute of Finance of Shanghai Jiao Tong University research group, this paper forecasts the regional GDP growth rate of China's power grids under the above three scenarios and discusses how the corresponding renewable energy quota system can be realized.

### 5.2. Data Sources

   In this paper, 2020 was selected as the base year for the research period 2021–2030, and the discount rate was set to 10%. Since China's water conservancy and power generation is very mature, annual power generation is relatively stable. In addition, the cycle of site selection and construction of nuclear power in the early stage is very long; the operational cost of a nuclear power plant is large; the regional distribution of nuclear power generation is uneven; and the change in power generation is small. The generation of other non-hydropower energy such as geothermal and tidal is relatively small, so it is not included in the calculation. Therefore, this paper does not consider hydropower and nuclear power generation technology but mainly research into the optimal allocation of installed non-hydropower capacity, including renewable energy such as wind, solar, and biomass.

   China has divided the power industry into six power grids according to geography, resource endowment, and transmission line structure: Northeast, North China, Northwest,

East China, Central China, and South China. As the main body responsible for implementing RPS, power grid enterprises are also the principal means to ensure the full acquisition of renewable energy. Therefore, this paper takes the six regional power grids as the research object to explore the dynamic distribution scheme of renewable energy power quota indicators. The installed power generation capacity of wind, solar, and biomass energy in the six regions in 2020 is shown on maps of current renewable energy (Figures 1–3).

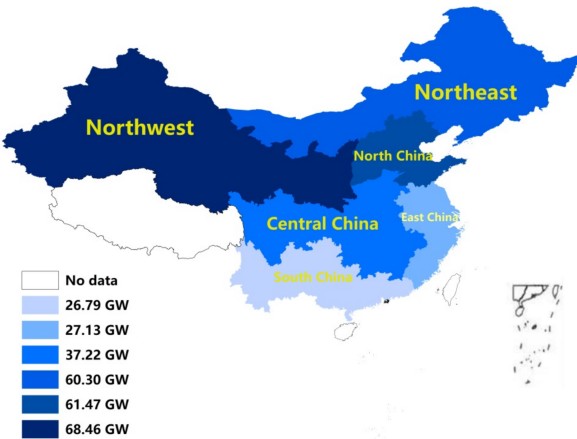

**Figure 1.** Regionally installed wind capacity in 2020.

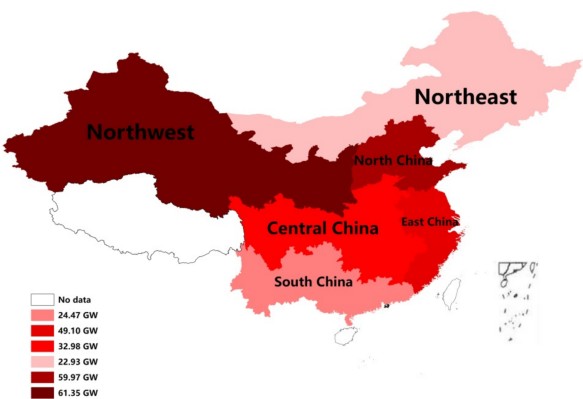

**Figure 2.** Regionally installed PV capacity in 2020.

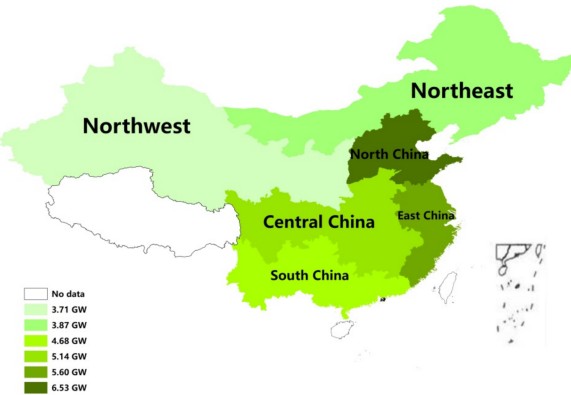

**Figure 3.** Regionally installed capacity of biomass in 2020.

The levelized cost of electricity (LCOE) is the average lifetime costs or long-run average costs. It represents the net discounted cost to install and operate an RE project divided by the power generation over its lifetime. The LCOE method has been widely used to evaluate power generation costs and the economic feasibility of electricity generation [58,59]. The

LCOE is the expected net present value of the unit cost of electricity over the lifetime of a generating asset. It can be classified into initial and annual costs (where the former is the one-time capital cost and the latter includes operations and maintenance costs), interest payments, and taxes. According to the learning curve theory, the initial cost can be calculated by the accumulated installed capacity, accumulated R&D funds, and the rated installed capacity. Based on the historical data of the above parameters, the LCOEs of all types of RE generation are calculated and used for corresponding predictions.

Based on the *China Electric Power Statistical Yearbook* over the years, this paper made a detailed investigation of the historical installed capacity of power production in each region, including the production type, installed capacity, and construction year of each region. Due to the difference in resource endowment, and the limited amount of non-fossil energy in each region, the annual power generation duration was also different. Assuming that the annual power generation duration of biomass was the same in each region (Cheng et al. [58]), it was 4412 h according to the national average level. Yi and Xu [59] gave specific data on the energy ceiling and the annual power generation duration of each type in the power grid area.

Due to the long distance between power grid areas, usually more than 600 km, high-voltage transmission was adopted, and direct current was more economical (Wang et al. [60]). Each region was regarded as a whole, so the line construction in the region was not within the scope of this paper. The transmission limit between power grids was determined according to the maximum capacity of transmission lines. Yi and Xu [59] estimated the transmission loss and cost according to the characteristics and distance of high-voltage lines based on an investigation of historical transmission lines between regions. The cost of a transmission line included two parts: one fixed (mainly infrastructure such as a substation) and the other variable related to the transmission distance. The specific data of the unit transmission cost and loss rate are shown in Table 1.

The proportion of RE generation in each power grid region varied greatly because of differences in regions and resources. The consumption proportion target for many years of non-hydropower renewable energy in each administrative region formulated by the National Energy Administration can be understood as the quota of each region in the RPS policy. According to the model established by Yi and Xu [59], the results of the proportion of non-hydropower renewable energy power generation in each region during the study period were obtained. According to the relevant historical data of the *China Electric Power Statistical Yearbook 2021*, the power generation capacity of coal and natural gas was predicted.

The technical flexibility coefficient represents the contribution of the power system to power generation. By optimizing the power structure, the flexibility requirements of power system can be met. Wind and solar power require additional system flexibility to smooth fluctuations, and its flexibility coefficient is negative (Sullivan et al. [61]). Gas and biomass power generation can quickly respond to power fluctuations. As a basic power source, coal-fired power plants do not adapt to frequent start-and-stop operations, so their flexibility coefficient is slightly low. Meanwhile, part of the power demand needs to be met by a flexible power supply.

## 6. Results and Discussion

### 6.1. Regional Renewable Energy Installation Optimization Results

Based on the above research assumptions and data sources, the optimal allocation model of regional renewable energy power generation was designed so that each power grid met the medium- and long-term objectives of the RPS by producing renewable energy power generation at the lowest cost. The improved genetic algorithm was used to solve the regional RE generation allocation optimization model, and the optimal allocation scheme of RE generation installed capacity under different scenarios was obtained.

Figures 4–6 are the numerical results of the cumulative installed capacity of different RE power generations under the three scenarios at the end of the study period. Meeting

the rapid growth of power demand and the target policy of the renewable energy quota system, wind and solar power generation will be the focus of non-hydropower renewable energy development. Wind power generation will develop rapidly in the short term, while solar power, with cost advantages in the medium and long term, will gradually accelerate its development with the reduction in cost. For various kinds of renewable energy, the installed power generation capacity of all technologies under different scenarios increases with time, and the growth trend is generally the same.

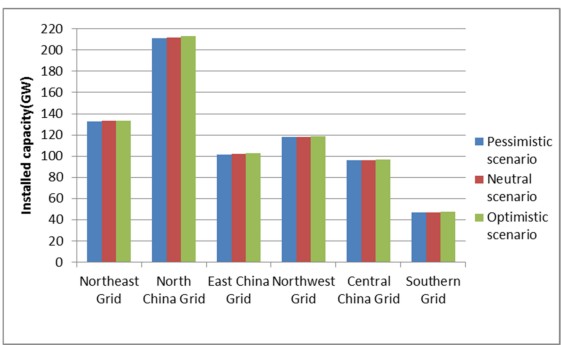

**Figure 4.** Regionally installed wind capacity under the three scenarios in 2030.

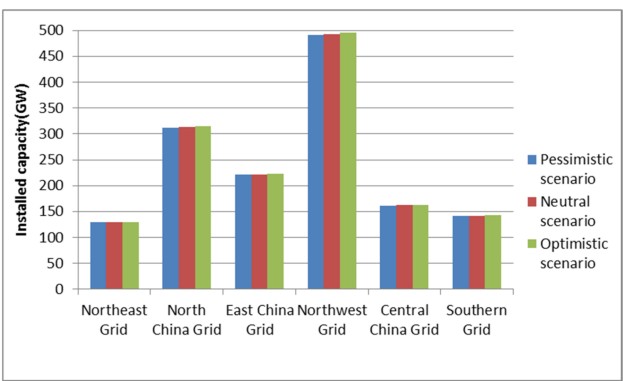

**Figure 5.** Regionally installed PV capacity under the three scenarios in 2030.

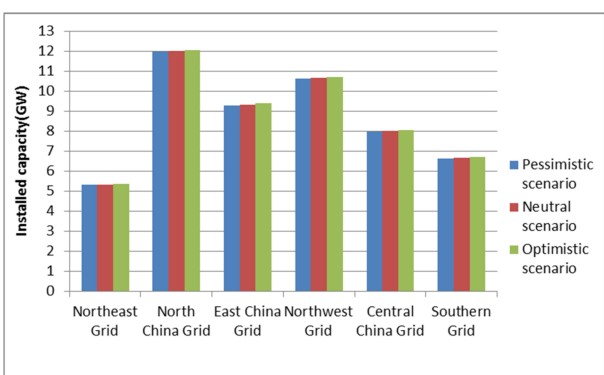

**Figure 6.** Regionally installed biomass capacity under the three scenarios in 2030.

From the numerical results of cumulative installed capacity in Figures 4–6, it can be seen that, by the end of 2030, the maximum installed capacity of non-hydropower RE power generation in China will be about 1667.39 GW, and the minimum will be about 1655.54. Among these, the North China Power Grid has the largest newly installed wind power generation capacity, while the Northwest Power Grid has the largest newly installed solar and biomass power generation capacity.

The optimal allocation results of the non-hydropower installed capacity of each power grid under the neutral scenario are given below. Figures 4–6 are the time series of installed capacity of wind energy, solar energy, and biomass energy under this scenario.

From 2021 to 2025, the installed capacity of wind power in the East China Power Grid increased the most; the installed capacity of solar power in the Northwest Power Grid increased the fastest; and the capacity of biomass power generation in the Northwest Power Grid increased the most. As can be seen from Figure 7, by the end of the 14th Five-Year Plan, the North China Power Grid had the largest installed wind power generation capacity, with the northwest, northeast, East China, Central China, and southern power grids ranking 2–6, respectively. The sum of the top-four grid capacities accounted for about 80% of total wind power generation. Figure 8 shows that the Northwest Power Grid accounted for the largest proportion of solar power generation, and the total installed capacity of the top-four grids in the northwest, North China, East China, and Central China accounted for more than 80% of the national installed capacity. According to the data in Figure 9, biomass power generation accounted for the largest proportion in the North China Power Grid, and the total installed capacity of biomass power generation in North China, East China, northwest, and Central China Power Grid accounted for about 75% of the total installed capacity. In general, due to the different locations of the regional grids, the Southern China Power Grid was less dependent on wind energy, and the solar energy and biomass energy in the northeast were relatively scarce in the medium and long-term renewable energy quota system.

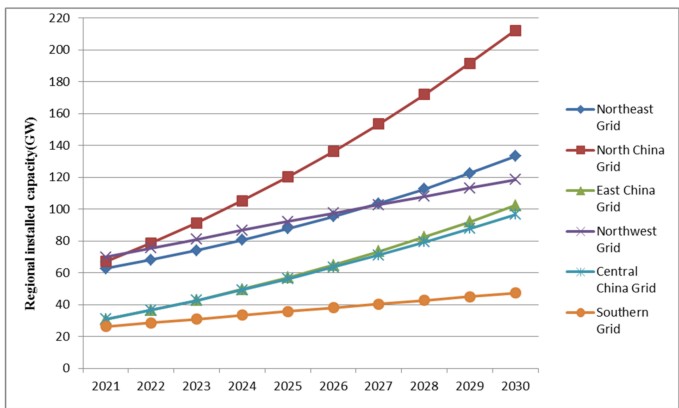

**Figure 7.** Installed capacity time series of wind under the neutral scenario.

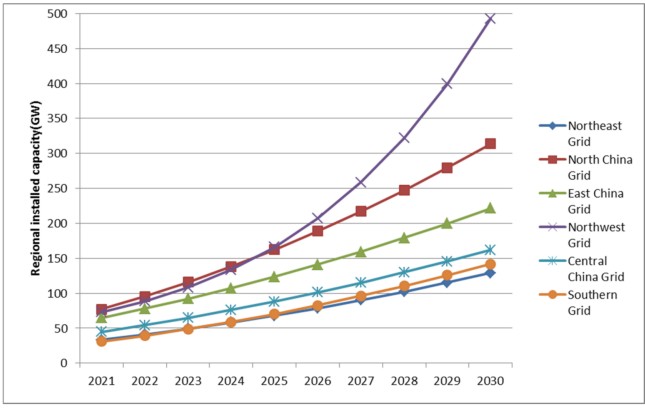

**Figure 8.** Installed capacity time series of PV under the neutral scenario.

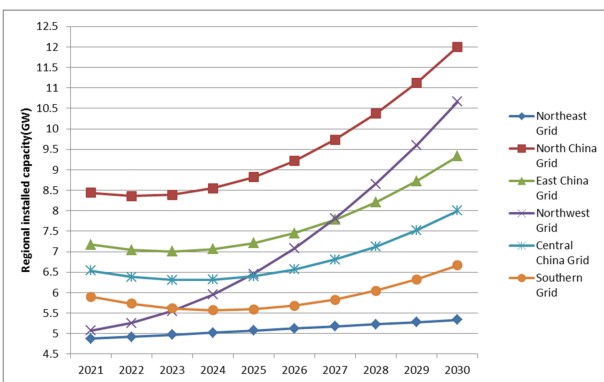

**Figure 9.** Installed capacity time series of biomass under the neutral scenario.

During China's next Five-Year Plan (2026–2030), the installed capacity of wind power generation in the North China Power Grid will increase significantly; the installed capacity of solar power generation in Northwest China Power Grid will increase the most, while biomass power generation will still be the largest in the Northwest. The data in Figures 4–6 show that, by the end of 2030, the installed capacity of wind power in the North China Power Grid will still account for a large proportion, and the sum of the installed capacity of wind power in North China, northeast, northwest, and East China will account for more than 80% of the national total. The Northwest Power Grid has the largest installed capacity of solar power generation. The total installed capacity of northwest, North China, and East China accounts for more than 70% of the national total. Biomass power generation accounts for the largest proportion in the North China Power Grid, and the total installed capacity of biomass power generation in North China, northwest, East China, and the Central China Power Grid accounts for more than 75% of total installed capacity. From a regional perspective, the total installed capacity of non-hydropower renewable energy in the Northwest China Power Grid is much higher than that of other regional power grids. On the one hand, Northwest China has the advantages of high altitude, vast territory, large air volume, ample sunshine, and huge resource potential. On the other hand, due to the low cost of renewable energy generation in the region, many cross-regional transmission lines were built between it and East China, Central China, and other grids during the study period to promote the consumption of renewable energy.

*6.2. Comparative Analysis of Different Scenarios*

This study considered three scenarios of the pandemic's impact on the economy: optimistic, neutral, and pessimistic. The newly installed capacity results of power generation such as wind, solar, and biomass energy under the three scenarios are shown in Figures 10–12. By comparing the new capacity of non-hydropower RE generation of all grids under various scenarios, the following conclusions can be drawn:

(1) By the end of 2030, under the pessimistic scenario, in which the pandemic cannot be effectively controlled, the newly installed capacity of non-hydropower RE power generation will be about 1655.54 GW, of which the largest newly installed generation capacity of wind power in the North China Power Grid will be about 150.14 GW. The region with the largest newly installed capacity of solar and biomass power generation will be the Northwest Power Grid, 429.98 and 6.93 GW, respectively.

(2) Under the neutral scenario from 2021 to 2030, the total newly installed capacity of non-hydropower RE power generation will be about 1661.01 GW, of which the newly installed capacity of wind power generation in the North China Power Grid will be the largest at about 150.62 GW. The top-two regions with newly installed solar power generation capacity are the Northwest and North China Power Grids, with new capacities of 431.57 and 253.52 GW, respectively. Regarding biomass power generation, the Northwest Power Grid will add 6.96 GW of installed capacity.

(3)  Under the optimistic scenario, in which the pandemic has been effectively controlled, the newly installed capacity of non-hydropower RE power generation during the study period will be about 1667.39 GW, and the total installed capacity will be at least 18.78 GW more than that under the pessimistic scenario. As can be seen from Figure 6, the region with the largest newly installed wind power generation capacity is the North China Power Grid, while the region with the largest newly installed solar and biomass power generation capacity is still the Northwest.

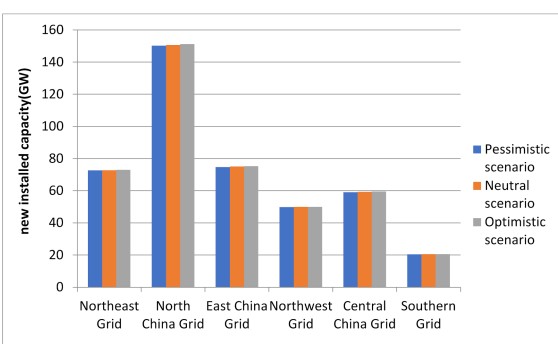

**Figure 10.** Newly installed wind power capacity under the three scenarios.

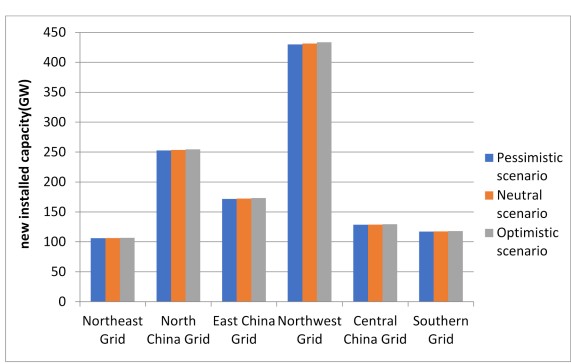

**Figure 11.** Newly installed PV power capacity under the three scenarios.

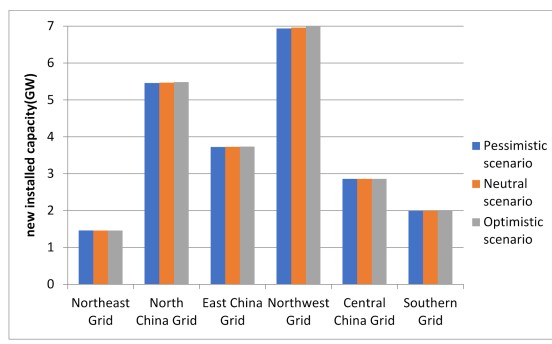

**Figure 12.** Newly installed biomass power capacity under the three scenarios.

## 7. Conclusions

The RPS is a constraint for regional power grids, and the quota policy determines the power production decision of each province to a certain extent. Combined with the medium- and long-term renewable energy generation quota objectives of different regions, this paper constructed a multi-regional power allocation optimization model and planned the RE power generation of each power grid. The main conclusions are the following:

(1)  Under the RPS system, the newly installed capacity of renewable energy will be mainly non-hydropower. By 2030, China's installed capacity of non-hydropower renewable energy will reach 2.2 TW. Therefore, we should make full use of existing

resources, explore new and clean energy, and decompose and arrange indicators for each province and city according to the allocation results of regional renewable energy, which can further optimize the energy layout of each region.

(2) The results of the optimal allocation model established in this paper show that in non-hydropower renewable energy, the installed capacity of wind power generation accounts for about 31.9% of total installed non-hydropower RE capacity, and the capacity of solar power generation accounts for 65.7% of total installed capacity. The implementation of the RPS policy is to encourage the redevelopment of renewable energy in various regions. Therefore, the effects of implementing it can be determined by comparing the installed power generation capacity under different scenarios of the pandemic.

Under the RPS, China's power grids take the energy structural adjustment as the primary task for meeting state-allocated quotas. At the same time, steady economic growth must be maintained and high production efficiency must be achieved in all regions. From a long-term perspective, all grids should maintain current wind and solar power generation, give priority to the development of wind resources in North China, the northeast, and other regions, to solar resources in Northwest China, and to biomass power generation in other regions to achieve a renewable energy quota. The implementation of RE policy promotes the construction of cross-regional transmission lines between resource-rich and resource-deficient areas. These will provide more channels for the outward transmission of clean energy to reduce cross-regional coal transportation.

**Author Contributions:** Conceptualization, Y.H. and L.W.; methodology, Y.H., M.Z., and H.Z.; data collection, L.W. and H.Z.; empirical analysis, Y.H. and H.Z.; writing—original draft preparation, Y.H., L.W., and H.Z.; writing—review and editing, Y.H. and H.Z.; supervision, H.Z. H.Z. is the corresponding author. All authors have read and agreed to the published version of the manuscript.

**Funding:** This research was funded by the Youth Science Foundation Program of NSFC, grant number 71701103, and was funded by Y.H.

**Institutional Review Board Statement:** Not applicable.

**Informed Consent Statement:** Not applicable.

**Data Availability Statement:** The data presented in this study can be downloaded by: https://pan.baidu.com/s/1DlfC3AOD5yGRzBKrUz5pIQ?pwd=1234 (accessed on 19 April 2022) with download password: 1234.

**Conflicts of Interest:** The authors declare no conflict of interest.

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
