# Peer review of "Regional Renewable Energy Installation Optimization Strategies with Renewable Portfolio Standards in China"

_sustainability, doi:10.3390/su141710498_

Round 1

Reviewer 1 Report

Overall well written and presented results, however, I only suggest to rewrite the conclusion to be shorter and only focus on the main outcomes of the current work. 

Author Response

Reviewer 1. Overall well written and presented results, however, I only suggest to rewrite the conclusion to be shorter and only focus on the main outcomes of the current work. 

Reply:

Dear reviewer, thanks for reviewing my manuscript during your busy schedule and for your affirmation and valuable comments. I refine the “conclusions” into “

(1)Under China's RPS system, due to the limitation of hydropower resources and policies, the IInewly installed capacity of renewable energy in China will be mainly non-hydropower in the future. By 2030, China's installed capacity of non hydropower renewable energy will reach 2.2tw. Therefore, we should make full use of existing resources, explore the potential of new and clean energy, decompose and arrange indicators for each province and city according to the allocation results of regional renewable energy, which can further optimize the energy layout of each region in China.

(2)The solution results of the optimal allocation model established in this paper show that in non-hydropower renewable energy, the installed capacity of wind power generation accounts for about 31.9% of the total installed capacity of non-hydropower RE, and the installed capacity of solar power generation accounts for 65.7% of the total installed capacity. The implementation of RPS policy is to encourage the redevelopment of renewable energy in various regions in China. Therefore, the implementation effect of RPS can be determined by comparing the installed power generation capacity under different scenarios of the epidemic.

Under the RPS, China's power grids take energy structural adjustment as the primary task to meet the quota indicators allocated by the state to each region. At the same time, we need to maintain the steady growth of economic development and achieve high production efficiency in all regions. From a long-term perspective, all grids should maintain the current development of wind power and solar power generation, give priority to the development of wind resources in North China, Northeast, and other regions, as well as solar resources in Northwest China, along with biomass power generation in power grids in other regions, so as to achieve the goal of renewable energy quota policy in all regions of China. The implementation of RE policy promotes the construction of cross regional transmission lines between resource rich areas and resource deficient areas. In the future, cross regional transmission lines will provide more channels for the outward transmission of clean energy, so as to reduce the cross regional coal transportation volume.”

Reviewer 2 Report

Dear authors. I enjoy reading your article and find it insightful.

Author Response

Reviewer 2. Dear authors. I enjoy reading your article and find it insightful.

Reply:

Dear reviewer, thanks for reviewing my manuscript during your busy schedule. And I am so happy to receive your positive comments.

Reviewer 3 Report

Dear authors,

1 – Equations 1 and 2 have the same variable LC.

2 – Concerning parameters in Eq.1 to 2, some additional data may help on assessing the results. For instance, as the paper consider a five-year plan, the predicted costs are welcome.

3 – The GA results for the three scenarios are too similar. Please, provide more explanation about the results.

4 – The GA performance is omitted from the results.

5 – Other GA parameters are also omitted. Population size, crossover parameters, maximum interactions.

6 – Moving SCENARIOS DESIGN (line 497) to an early part may help. GDP rates (line 506) used are also welcome.

Author Response

Reviewer 3

Reply:

Dear reviewer, thanks for reviewing my manuscript during your busy schedule and for your detailed comments and sincere suggestions.

1 – Equations 1 and 2 have the same variable LC.

Reply 1.  Change equation 2 to

2 – Concerning parameters in Eq.1 to 2, some additional data may help on assessing the results. For instance, as the paper consider a five-year plan, the predicted costs are welcome.

Reply 2. According to your suggestion, the paper revised the power generation cost into two parts: initial investment cost and annual cost. Among them, the initial cost is the one-time investment cost of the equipment, and the annual cost includes operation cost, maintenance cost, interest payment and taxes.

3 – The GA results for the three scenarios are too similar. Please, provide more explanation about the results.

Reply 3. This paper designs three scenarios: optimistic scenario, neutral scenario and pessimistic scenario. The difference between different scenarios is mainly the difference in GDP growth rate. According to the calculation of global GDP growth by the research group of China Institute of finance, Shanghai Jiaotong University, the difference of regional GDP growth rate of China's power grids under the three scenarios is not great, so the numerical results of RE power generation installed capacity under different scenarios are relatively close. However, the final RE installed capacity is different. Add the following contents after Fig. 6:

“From the numerical results of cumulative installed capacity in Figure 1-3, it can be seen that, by the end of 2030, the maximum installed capacity of non-hydropower RE power generation in China will be about 1667.39gw, and the minimum will be about 1655.54gw affected by the current situation and development of the global epidemic. Among them, the North China Power Grid has the largest newly installed capacity of wind power generation, while the Northwest Power Grid has the largest newly installed capacity of solar and biomass power generation.”

4 – The GA performance is omitted from the results.

5 – Other GA parameters are also omitted. Population size, crossover parameters, maximum interactions.

Reply 4 and 5. According to the above two modification suggestions, the GA parameters and GA performance analysis are added after Article 4.2:

“4.3 Algorithm Performance Analysis

Before solving the generation allocation optimization model, set the parameters for the genetic algorithm: the initial population size is 500, the crossover probability is 0.6, the mutation probability is 0.02, and the maximum evolution number is 250. The working software is python3.7 and the computing environment is Intel (R) core (TM) i7_ 7500u CPU @ 2.70 GHz 2.90ghz, 16GB operating memory, and win10 operating system.

Analyze the solution effect of the improved genetic algorithm. Table 1 covers the the algorithm performance of calculating the optimal target value after normalization, showing that the target mean value, optimal value, worst value and average deviation are better when IgA solves the RE power distribution problem. The performance analysis shows that IGA can not only ensure the convergence speed, but also improve the quality of solutions when solving multi-objective problems.

Table1 Algorithm Performance Analysis for Solving Optimization Problems

algorithm

target mean value

optimal value

worst value

average deviation

IGA

0.278

0.214

0.354

0.020

"

6 – Moving SCENARIOS DESIGN (line 497) to an early part may help. GDP rates (line 506) used are also welcome.

Reply 6.

According to your review comments, moving the “SCENARIOS DESIGN” to the front of “Data Sources” and the “GDP rate” to the front of “Data Sources”.

Reviewer 4 Report

The author could include the parameters for policy decision after conducting an extensive survey from the stake holders rather than choosing the parameters as a thumb rule. 

Author Response

Reviewer 5.The article is very interesting. However, you should consider answering the questions and making some additions.

Reply:

Dear reviewer, thanks for reviewing my manuscript during your busy schedule and for your detailed comments and sincere suggestions.

Comment 1: The RPS follows the key introduction by China’s NDRC, the Ministry of Finance, and the NEA of the "531" policy, which sought to control the surge of PV by phasing out subsidies. So what were the premises for analyzing the development of PV in the conditions of inhibiting the development of RES? 

Reply 1:

During the "fourteenth five year plan" period, China will anchor the goal of carbon peak and carbon neutrality, give full play to the resource and technological advantages of China's renewable energy, consolidate and enhance the overall core competitiveness of renewable energy, and promote the market-oriented development of high-quality and low-cost renewable energy.

Accelerating the construction of a new power system with new energy as the main body to improve the consumption and storage capacity of new energy. It can not only realize the large-scale development of renewable energy, but also achieve a high level of consumption and utilization, more effectively ensuring the reliable and stable supply of power, and achieving high-quality leapfrog development.

The decisive role of the market in the allocation of renewable energy resources will be further brought into play. From this year, the development of wind power PV will enter the parity stage, get rid of the dependence on financial subsidies, and achieve market-oriented and competitive development.

Comement 2: At the same time, the new policy creates challenges for coal-fired energy producers, as increased competition from RES companies may result in a decline in use and lower average tariffs. How does this situation translate into the implementation of RPS targets?

Reply 2:

The new policy has brought challenges to coal-fired energy producers. In this case, enterprises should be encouraged to make technological innovation. At the same time, due to regional differences, cancelling electricity price subsidies step by step to achieve stable and orderly access to the Internet at parity. In addition, taking full account of the development potential of regional resources and the lock-in effect of power generation technology, gradually solicit carbon taxes with high tax rates. Meanwhile, taking into account the effect and cost of emission reduction, set long-term development goals of renewable energy quota policy, promoting the implementation of the mandatory quota system and producing a marked effect as soon as possible.

Comement 3: Costs are an important element of investments. The authors of the article point to equation (1) (line 339), which is used to calculate the power generation cost of wind energy, solar energy, and other renewable energy sources. You should consider adding information about the cost level of individual investments based on a literature review or an investment cost analysis. The analysis would indicate information at what level of prices and subsidies it is profitable to invest in RES.

Reply 3:

Delete “The main connection between power production and power transmission is that they work together to meet the power demand of each region. Therefore, the total cost of RE power during the study period includes generation cost and transmission cost. The Levelized Cost of Energy (LCOE) includes the cost of investment, operation and maintenance of power production technology. Hence Eq. (1) is used to calculate the power generation cost of wind energy, solar energy and other renewable energy sources.                               (1)”

Add the following content, Power generation cost mainly includes initial investment cost and annual cost. Among them, the initial cost is the one-time investment cost of the equipment, and the annual cost includes operation cost, maintenance cost, interest payment and taxes [1]. According to the learning curve theory, the equipment production cost decreases with the increase of equipment output and R&D funds [2]. Therefore, the unit investment cost  based on the two factor learning curve can be expressed as:

Where, refers to the unit investment cost of energy  in the base year; refers to the cumulative installed capacity;  refers to the accumulated R&D funds;  refers the learning rate index of , and  refers the value range of , the value range of  and  is [0,1];

The levelized power cost and annual cost of renewable power are respectively expressed as:

Where,  indicates the type of renewable energy;  indicates the feed-in tariff of renewable energy ;  indicates the life cycle of the equipment ;  indicates the rated capacity of the equipment ;  indicates the actual operating hours of the equipment  in the year ;   indicates the discount rate;   indicates operation and maintenance costs;  indicates interest expense;  indicates taxes.”

Comement 4: Lines 440-442 ‘China has divided the power industry into regions…’ It would be worth adding a map showing the conditions for the development of individual renewable energy sources (map of insolation, winds, biomass) in the analyzed regions. The visual presentation is more appealing to the readers.

Reply 4:

In this paper, the installed power generation capacity of wind, solar and biomass energy in six grid regions in 2020 is drawn into a map of the current situation of renewable energy, as shown in Figure 1-3.

Comment 5: Do authors have access to data from before 2020? The analysis would give an insight into the situation before the Covid outbreak.

Reply 5:

Many parameters in the model constructed in this paper, including power production cost, power transmission cost, resource endowment and installed capacity of various types of technologies in each region, which have taken into account the historical data of China's power Yearbook from 2011 to 2020.

Comment 6: The literature review can be supplemented (f.e. Jing-LiFan, et. al.; 2021; Shangjia Wang, et al. 2022).

Reply 6:Added the literature “Fan, J., Wang Q., and Zhang X. (2021), “A bibliometric analysis of the water-energy-food nexus based on the SCIE and SSCI database of the Web of Science”, Mitigation and Adaptation Strategies for Global Change, No. 8, 26.”

Reviewer 5 Report

The article is very interesting. However, you should consider answering the questions and making some additions.

The RPS follows the key introduction by China’s NDRC, the Ministry of Finance, and the NEA of the "531" policy, which sought to control the surge of PV by phasing out subsidies. So what were the premises for analyzing the development of PV in the conditions of inhibiting the development of RES? 

At the same time, the new policy creates challenges for coal-fired energy producers, as increased competition from RES companies may result in a decline in use and lower average tariffs. How does this situation translate into the implementation of RPS targets?

Costs are an important element of investments. The authors of the article point to equation (1) (line 339), which is used to calculate the power generation cost of wind energy, solar energy, and other renewable energy sources. You should consider adding information about the cost level of individual investments based on a literature review or an investment cost analysis. The analysis would indicate information at what level of prices and subsidies it is profitable to invest in RES.

Lines 440-442 ‘China has divided the power industry into regions…’ It would be worth adding a map showing the conditions for the development of individual renewable energy sources (map of insolation, winds, biomass) in the analyzed regions. The visual presentation is more appealing to the readers.

Do authors have access to data from before 2020? The analysis would give an insight into the situation before the Covid outbreak.

The literature review can be supplemented (f.e. Jing-LiFan, et. al.; 2021; Shangjia Wang, et al. 2022).

Author Response

Reviewer 5.The article is very interesting. However, you should consider answering the questions and making some additions.

The RPS follows the key introduction by China’s NDRC, the Ministry of Finance, and the NEA of the "531" policy, which sought to control the surge of PV by phasing out subsidies. So what were the premises for analyzing the development of PV in the conditions of inhibiting the development of RES? 

At the same time, the new policy creates challenges for coal-fired energy producers, as increased competition from RES companies may result in a decline in use and lower average tariffs. How does this situation translate into the implementation of RPS targets?

Costs are an important element of investments. The authors of the article point to equation (1) (line 339), which is used to calculate the power generation cost of wind energy, solar energy, and other renewable energy sources. You should consider adding information about the cost level of individual investments based on a literature review or an investment cost analysis. The analysis would indicate information at what level of prices and subsidies it is profitable to invest in RES.

Lines 440-442 ‘China has divided the power industry into regions…’ It would be worth adding a map showing the conditions for the development of individual renewable energy sources (map of insolation, winds, biomass) in the analyzed regions. The visual presentation is more appealing to the readers.

Do authors have access to data from before 2020? The analysis would give an insight into the situation before the Covid outbreak.

The literature review can be supplemented (f.e. Jing-LiFan, et. al.; 2021; Shangjia Wang, et al. 2022).

Reply:

Dear reviewer, thanks for reviewing my manuscript during your busy schedule and for your detailed comments and sincere suggestions.

Comment 1: The RPS follows the key introduction by China’s NDRC, the Ministry of Finance, and the NEA of the "531" policy, which sought to control the surge of PV by phasing out subsidies. So what were the premises for analyzing the development of PV in the conditions of inhibiting the development of RES? 

Reply 1:

During the "fourteenth five year plan" period, China will anchor the goal of carbon peak and carbon neutrality, give full play to the resource and technological advantages of China's renewable energy, consolidate and enhance the overall core competitiveness of renewable energy, and promote the market-oriented development of high-quality and low-cost renewable energy.

Accelerating the construction of a new power system with new energy as the main body to improve the consumption and storage capacity of new energy. It can not only realize the large-scale development of renewable energy, but also achieve a high level of consumption and utilization, more effectively ensuring the reliable and stable supply of power, and achieving high-quality leapfrog development.

The decisive role of the market in the allocation of renewable energy resources will be further brought into play. From this year, the development of wind power PV will enter the parity stage, get rid of the dependence on financial subsidies, and achieve market-oriented and competitive development.

Comement 2: At the same time, the new policy creates challenges for coal-fired energy producers, as increased competition from RES companies may result in a decline in use and lower average tariffs. How does this situation translate into the implementation of RPS targets?

Reply 2:

The new policy has brought challenges to coal-fired energy producers. In this case, enterprises should be encouraged to make technological innovation. At the same time, due to regional differences, cancelling electricity price subsidies step by step to achieve stable and orderly access to the Internet at parity. In addition, taking full account of the development potential of regional resources and the lock-in effect of power generation technology, gradually solicit carbon taxes with high tax rates. Meanwhile, taking into account the effect and cost of emission reduction, set long-term development goals of renewable energy quota policy, promoting the implementation of the mandatory quota system and producing a marked effect as soon as possible.

Comement 3: Costs are an important element of investments. The authors of the article point to equation (1) (line 339), which is used to calculate the power generation cost of wind energy, solar energy, and other renewable energy sources. You should consider adding information about the cost level of individual investments based on a literature review or an investment cost analysis. The analysis would indicate information at what level of prices and subsidies it is profitable to invest in RES.

Reply 3:

Delete “The main connection between power production and power transmission is that they work together to meet the power demand of each region. Therefore, the total cost of RE power during the study period includes generation cost and transmission cost. The Levelized Cost of Energy (LCOE) includes the cost of investment, operation and maintenance of power production technology. Hence Eq. (1) is used to calculate the power generation cost of wind energy, solar energy and other renewable energy sources.                               (1)”

Add the following content, Power generation cost mainly includes initial investment cost and annual cost. Among them, the initial cost is the one-time investment cost of the equipment, and the annual cost includes operation cost, maintenance cost, interest payment and taxes [1]. According to the learning curve theory, the equipment production cost decreases with the increase of equipment output and R&D funds [2]. Therefore, the unit investment cost  based on the two factor learning curve can be expressed as:

Where, refers to the unit investment cost of energy  in the base year; refers to the cumulative installed capacity;  refers to the accumulated R&D funds;  refers the learning rate index of , and  refers the value range of , the value range of  and  is [0,1];

The levelized power cost and annual cost of renewable power are respectively expressed as:

Where,  indicates the type of renewable energy;  indicates the feed-in tariff of renewable energy ;  indicates the life cycle of the equipment ;  indicates the rated capacity of the equipment ;  indicates the actual operating hours of the equipment  in the year ;   indicates the discount rate;   indicates operation and maintenance costs;  indicates interest expense;  indicates taxes.”

Comement 4: Lines 440-442 ‘China has divided the power industry into regions…’ It would be worth adding a map showing the conditions for the development of individual renewable energy sources (map of insolation, winds, biomass) in the analyzed regions. The visual presentation is more appealing to the readers.

Reply 4:

In this paper, the installed power generation capacity of wind, solar and biomass energy in six grid regions in 2020 is drawn into a map of the current situation of renewable energy, as shown in Figure 1-3.

Comment 5: Do authors have access to data from before 2020? The analysis would give an insight into the situation before the Covid outbreak.

Reply 5:

Many parameters in the model constructed in this paper, including power production cost, power transmission cost, resource endowment and installed capacity of various types of technologies in each region, which have taken into account the historical data of China's power Yearbook from 2011 to 2020.

Comment 6: The literature review can be supplemented (f.e. Jing-LiFan, et. al.; 2021; Shangjia Wang, et al. 2022).

Reply 6:Added the literature “Fan, J., Wang Q., and Zhang X. (2021), “A bibliometric analysis of the water-energy-food nexus based on the SCIE and SSCI database of the Web of Science”, Mitigation and Adaptation Strategies for Global Change, No. 8, 26.”

Round 2

Reviewer 3 Report

No more suggestions.